# Inspired by the Nature: A Post-printed Strategy to Efficiently Elaborate Parahydrophobic Surfaces

**DOI:** 10.3390/biomimetics7030122

**Published:** 2022-08-28

**Authors:** Jordy Queiros Campos, Caroline R. Szczepanski, Marie Gabrielle Medici, Guilhem Godeau

**Affiliations:** 1Université Côte d’Azur, INPHYNI, UMR 7010, 06000 Nice, France; 2Department of Chemical Engineering & Materials Science, Michigan State University, East Lansing, MI 48824, USA; 3Université Côte d’Azur, IMREDD, 06200 Nice, France

**Keywords:** plants, bioinspiration, parahydrophobic, harvesting, 3D printing, post-functionalization

## Abstract

The lack of drinkable water is one of the most significant risks for the future of the humanity. Estimates show that in the near future, this risk will become the origin of massive migrations leading to humanitarian disaster. As consequence, the development of solutions to provide water is becoming ever more critical, and a significant effort is devoted to identifying new sources of water. Among the developed strategies, fog harvesting, which takes advantage of atmospheric water to provide potable water, is a solution of interest due to its potential in sustainable development. Unfortunately, this approach suffers from low yield. In the present work, we take inspiration from living species to design and elaborate surfaces with high potential for water harvesting applications. This work takes advantage of 3D-printing and post-printing functionalization to elaborate a strategy that allows modelling, printing, and functionalization of surfaces to yield parahydrophobic behavior. The roughness and surface morphology of the prepared surfaces were investigated. These characteristics were then related to the observed wettability and potential of the functionalized interfaces for water harvesting applications. This work highlights significant variations in surface wettability via surface modification; strong hydrophobic behavior was observed via modification with linear carboxylic acids particularly for surfaces bearing vertical blades (plate with vertical blades and grid with vertical blades).

## 1. Introduction

Climate change has many consequences for living species. Currently, one of the most significant problems associated with the evolving climate and environment is the associated decrease in drinkable water resources. Recent analyses estimate that by 2025, a significant portion of the global population will suffer from a lack of water [1]. Since water is vital for human life, this projected scenario could lead to a catastrophic humanitarian situation, with consequences of massive population migration and associated conflicts.

Some creative solutions have been identified which provide potable water to regions most impacted by this growing challenge. One of the more efficient strategies developed has been the desalinization of sea water [2,3]. This approach is employed in Saudi Arabia, mainly to generate water resources for irrigation. While effective, this approach ultimately suffers from severe drawbacks. First, this approach can only be developed in coastal environments due to the need for a significant volume of sea water. Second, the consequence of desalinating a large quantity of water is that it produces a huge amount of salt, creating hyper-saline regions and salt deserts that sterilize the surrounding area [4,5]. For this reason, new solutions are still being actively investigated and pursued globally. Among the alternative solutions, fog harvesting is of particular interest [6]. The concept of this strategy is to accumulate water microdroplets present in the atmosphere. This strategy is particularly interesting in desert countries with high hygrometric levels, such as Chile where fog harvesting is already commonly used for irrigation [7]. Another advantage of this strategy is that is has an associated compatibility with sustainable development. Unfortunately, harvesting yield is strongly linked with harvesting devices (mostly meshes) and environmental conditions. As consequence, this strategy may still be improvement to enable widespread implementation [8,9,10].

One approach to improve the yield and effectiveness of this strategy is to find inspiration in nature [11]. It is well known that there are animal and plant species that have developed techniques to manage water use and collection in regions where liquid water is scarce. As one example, the Namib desert beetle (Onymacris unguicularis) traps water on its legs and body [12,13,14,15,16]. Similarly, cacti can collect water with their spines and hairy plants can condense dew from the air to collect water and decrease transpiration [17,18,19,20,21].

Considering cacti examples, the capacity to harvest water is correlated with two parameters. The first of those parameters is the macrostructures present on each plant that increases the surface area available for water harvesting, mainly spines for cacti. These structures increase the specific surface area available for water harvesting and as consequence increase the overall harvesting yield. The second parameter is the wettability. Cacti spines have strong hydrophobic character and high water adhesion, this combination of properties is known as the petal effect or parahydrophobicity [22,23,24]. 

These two parameters contribute to the overall mechanism of fog harvesting, which can be briefly described as follows. Due to the high affinity between the surface of the spines and water, a water droplet will accumulate on these structures. As the volume of the water droplet becomes significantly large, the adhesion strength becomes too weak and the droplet can no longer be supported and is driven along the spines due to parallel grove on the spines that allow a directional collection [19]. Considering this observation from the nature, we hypothesize that both macroscopic morphology and surface wettability (parahydrophobicity) need to be simultaneously controlled to obtain surfaces capable of effective fog harvesting. 

In this work, we propose using a platform that combines strategies which separately control surface morphology and wettability. First, to control macroscopic surface morphology, we employ three-dimensional printing (3D-printing). 3D-printing is a technology with significant potential in a wide range of emerging fields including engineering with conductive polymers, desalination, and solar evaporators [25,26,27,28,29,30]. Furthermore, 3D printing is an attractive approach as it allows surfaces to be printed with a high accuracy. The surface wettability is ultimately more difficult to control for the intended work. It is well established in the research literature that surface wettability depends on the surface composition (surface energy) and the surface roughness (micro/nanos-scale morphology), with various strategies described to control both roughness and energy [31,32,33,34,35,36,37,38,39]. Unfortunately, in most cases the described strategies are used with metallic materials or conductive polymers and are thus difficult to adapt in order to modify 3D-printed surfaces. The present work establishes a strategy to both develop and functionalize post-printing surfaces with various designs (Figure 1). We employ a commercially available composite filament, comprised of 80% colloidal copper particles dispersed in poly-lactic acid for 3D-printing. The morphologies investigated here were designed to increase the specific surface area compared to a simple planar substrate. For post-printing functionalization, we take advantage of a previously described strategy where metallic copper is functionalized using a two-step approach [40,41]. Functionalization with a wide variety of carboxylic acids was investigated, and the influence of this functionalization on surface properties (wettability) and morphology are described.

## 2. Materials and Methods

### 2.1. Surface Elaboration

#### 2.1.1. Surface Modelling

The surfaces were modelled using Fusion 360 from Autodesk (San Rafael, CA, USA).

Plate with vertical blade (PVB): PVB was designed as a 2.5 × 2.5 × 2 cm square plate decorated with vertical blades of 3 mm height, 0.65 mm width, and separated by space of 1.3 mm.

Grid with vertical blade (GVB): GVB was designed as a 2.5 × 2.5 × 2.5 cm square grid with vertical blades of 0.62 mm width separated by space of 1.25 mm.

Large mesh grid (LMG): LMG was designed as a 2.5 × 2.5 × 0.25 cm square grid with mesh of 2.5 mm.

Narrow mesh grid (NMG): NMG was designed as a 2.5 × 2.5 × 0.25 cm square grid with mesh of 1 mm.

#### 2.1.2. Surface Printing

Surfaces were printed using a 3D-printer (Stream 30 pro MK 2, Volumic 3D, Nice, France). The filament used for printing was Cuivre 80 Ultra (Volumic Ultra). This filament is made of polylactic acid (PLA) and loaded with 80% of colloidal copper. The nozzle used for printing is a 25 µm brass nozzle. Extrusion temperature was 195 °C and reception plate temperature was 60 °C. 

### 2.2. Surface Functionalization

Step 1: The surfaces were immersed in a solution of NaOH (2.5 M) and (NH_4_)_2_S_2_O_8_ (0.13 M) and then placed on a shaker at room temperature for 2 h. During this period, the colorless solution became blue, confirming the release of Cu^2+^ ions. After this period, the surfaces were washed three times in water and once in ethanol. After washing, the surfaces were directly used for the next step without drying.

Step 2: The oxidized surfaces were immersed in carboxylic acid solution in ethanol (10 mg·mL^−1^) and slowly agitated at room temperature for 2 h. After this period, the surfaces were washed three times in ethanol. The surfaces were then dried in an oven overnight at 70 °C.

### 2.3. Surface Roughness

Roughness measurements were performed using an optical profilometer (InfiniteFocus G5 plus, Brucker Alicona, Graz, Austria. All measurements were performed three times on three different samples to calculate standard deviation.

### 2.4. Surface Morphologies

SEM observations were carried out using Phenom Pro X and Phenom XL Desktop SEM (Thermo Fisher Scientific, Waltham, MA, USA). Samples were observed with gold coating and at an accelerating voltage of 10 kV. The samples were coated with gold using Quorum Q150R S Sputter Coater (Quorum Technologies, Lewes, UK).

### 2.5. Surface Wettability

Contact angle measurements were performed on non-functionalized and post-functionalized surfaces. All measurements were performed 5 times to determine standard deviation. The apparent contact angles were obtained with a contact angle system OCA TBU100 (dataphysics). The apparent contact angles were measured using the sessile drop method with deionized water (droplet volume of 2 µL). 

## 3. Results

The goal of this research is to prepare parahydrophobic surfaces with various macroscopic morphologies designed to increase specific surface area and improve the water harvesting capability of the interfaces. To reach this goal, we employed 3D-printing technology. This approach allows us to print surfaces suitable for functionalization with a precise morphology.

### 3.1. Surface Elaboration

The first step is to design, and 3D print a surface that can be post-functionalized. Taking inspiration from nature, we chose surfaces with macroscopic substructures that increase the specific surface area and therefore increase the water harvesting capability of the surface. Four surfaces were designed as plates and grids, specifically a plate with vertical blades (PVB) (Figure 2A), a grid with vertical blades (GVB, Figure 2B) and a large and a narrow mesh grid (NMG and LMG, Figure 2C and Figure 2D, respectively). 

Since the printed surfaces need to be suitable for post-functionalization, these designs were printed from a commercially available composite filament comprised of 80% colloidal copper and poly-lactic acid. The printed surfaces are presented in Figure 3.

It is well reported in the literature that copper can be used for post-functionalization for various applications including the control of surface wettability [6,40,42,43]. In our group, we recently demonstrated that this approach was suitable for post-printing functionalization of surfaces [44].

### 3.2. Surface Functionalization

The printed surfaces (reported as Cu surface) were functionalized with various carboxylic acids following a two-step strategy. In the first step, copper is oxidized from metallic copper to Cu(OH)_2_ via immersion in a solution of ammonium persulfate (0.013 M) and sodium hydroxide (0.25 M) for two hours and at room temperature. The oxidized surface reported as Cu(OH)_2_ was then washed two times in water and once in ethanol. For the second step, the Cu(OH)_2_ surface was immersed in the functionalization solution for two hours at room temperature without further treatment. The functionalization solution contained 10 mg/mL of carboxylic acid in absolute ethanol. A wide variety of carboxylic acids were employed in this work (Figure 4) to evaluate the impact of the chemical modification on the final surface wettability.

To engineer hydrophobic properties, carboxylic acids with linear and non-polar chains were selected. The carboxylic acids were chosen with a wide variety of chain lengths to explore the impact of the side chain length on the final surface properties (Acetic acid (C_2_), hexanoic acid (C_6_), decanoic acid (C_10_), myristic acid (C_14_) and stearic acid (C_18_)). As negative controls, carboxylic acids with polar side chains and an associated lower hydrophobicity were considered (e.g., 2-(2-methoxyethoxy)acetic acid and glycolic acid, Methoxy and Gly, respectively). Finally, due to their low associated surface energy, fluorinated and perfluorinated carboxylic acids were also employed for functionalization (e.g., 4,4,5,5,6,6,7,7,8,8,9,9,9-tridecafluorononanoic acid, perfluoro pentanoic acid and perfluoro heptanoic acid, C_2_F_4_, F_4_ and F_6_ surfaces, respectively). After functionalization, all functionalized surfaces were rinsed three times in ethanol and then dried 24 h in oven at 70 °C prior to further analysis and characterization.

### 3.3. Surfaces Roughness

In a previous work, we characterized roughness on post-printed, functionalized plates [44]. On a plate, the majority of observed roughness resulted from the printing resolution. In this work, to obtain more accurate observations, roughness measurements were performed in the printing direction. This way, the impact of the printing resolution is less significant and allows us to observe variations relevant to the surface functionalization. Roughness measurements were performed on all the prepared surfaces and are presented in Figure 5.

The roughness can be compared functionalization by functionalization for the same design or design by design for the same functionalization. In the case of surfaces with substructures such as meshes or plates with blades, due to the surface macroscopic structures (blades or meshes) the roughness measurements are performed on the edge of the macroscopic structures. Here, the more interesting point is not the variations in roughness between the different functionalization but more the fact that for all designs a significant increase in roughness is observed after oxidation of the Cu surfaces confirming the impact of the oxidative treatment on the PLA. Raw copper surfaces present Ra of 2.8 ± 0.3 µm, 3.0 ± 0.5 µm, 7.4 ± 0.9 µm and 4.0 ± 1.4 µm, respectively, for PVB-Cu, GVB-Cu, LMG-Cu and NMG-Cu surfaces. Whereas oxidized surfaces present Ra of 9.8 ± 1.7 µm, 7.8 ± 1.1 µm, 11.3 ± 1.3 µm and 13.8 ± 2.4 µm, respectively, for PVB-Cu(OH)_2_, GVB-Cu(OH)_2_, LMG-Cu(OH)_2_ and NMG-Cu(OH)_2_ surfaces. Those Ra variations can be linked to changes in morphologies due to swelling when the surfaces are immersed in solution or contraction that occurs when the surfaces are dried. Additionally, the aggressive conditions used to convert Cu to Cu(OH)_2_ may degrade PLA (saponification), particularly at the edge of the blades or meshes. The partial saponification of PLA during oxidation has already been reported in the literature and can be confirmed by simple IR observation (Appendix A) [44]. For the same surface design, the differences in roughness depending on the functionalization may be significant, but it remains in the same range for most of the cases (all numerical data presented as supplementary data).

### 3.4. Surface Morphology

To confirm the observations made on roughness, scanning electronic microscopy was also performed (Figure 6).

These observations show that at low magnification, the impact of the oxidative treatment can be confirmed. Non-modified printed Cu surfaces (Figure 6A) present a homogenous surface where the copper particles are totally embedded into the PLA material. After oxidation, the Cu(OH)_2_ surface (Figure 6B) exhibit a rougher surface that releases the copper particles and is consistent with the evolution of the surface roughness (Ra) prior to and after oxidation. This is particularly interesting, since the degradation of the PLA surface exposes the copper particles and increases the surface available for functionalization. For functionalized surfaces, no significant difference can be observed compared with the oxidized Cu(OH)_2_ surface (Figure 6C,D). This observation is consistent with the literature [45,46]. With higher magnification, more significant modifications can be observed (Figure 7).

While the Cu surface appears to be quite smooth (Figure 7A), it is clearly observed that after oxidation (Figure 7B) and functionalization (Figure 7C,D), all surfaces present thin nano-crystallization. This crystal formation is consistent with the literature, which describes this kind of functionalization on copper surfaces. Obviously, the crystallization does not depend on the macroscopic shape of the surface and does not change from one design to the other as shown in Figure 8.

### 3.5. Surface Wettability

All surfaces were investigated for their wettability, as reported in Figure 9.

Similar to the roughness data, the wettability can be compared functionalization by functionalization for the same design or design by design for the same functionalization. Initially, Cu surfaces have hydrophobic behavior with apparent contact angles (*θ*) between 120° and 140° depending on the surface design (Figure 10A). Respectively, the contact angles are *θ* = 139.4° ± 0.4, *θ* = 138.9° ± 0.3, *θ* = 130.4° ± 2.0 and *θ* = 122.9° ± 0.4 for PVB Cu, GVB Cu, LMG Cu and NMG Cu surfaces.

Surprisingly, Cu(OH)_2_ surfaces present globally higher hydrophobic features compared with the Cu surfaces of the same design (PVB Cu(OH)_2_: *θ* = 146.2° ± 2.8, LMG Cu(OH)_2_: *θ* = 139.3° ± 1.3 and NMG Cu(OH)_2_: *θ* = 144.7° ± 1.0). Only the GVB Cu(OH)_2_ surface presents a lower contact angle value, but still remains strongly hydrophobic with *θ* = 128.9° ± 1.2. For surfaces functionalized with linear carboxylic acids, an increase in hydrophobicity is observed after functionalization. No significant differences are observed between the different linear carboxylic acids employed, and globally all carboxylic acid functionalization display strong hydrophobic properties. Only surfaces functionalized with acetic acid were “less” hydrophobic than the others, having apparent contact angles (*θ)* between 145° and 155°. Respectively, the contact angles are *θ* = 150.1° ± 0.4, *θ* = 154.4° ± 1.1, *θ* = 144.0° ± 1.2 and *θ* = 149.6° ± 2.0 for PVB C_2_, GVB C_2_, LMG C_2_ and NMG C_2_ surfaces. All other surfaces functionalized with C6, C10, C14 and C18 present contact angles near 155° or even higher (Figure 10B). All values are presented in Table 1.

Globally, the fluorinated and perfluorinated chains (C_2_F_4_, F_4_, F_6_) present lower contact angle values compared to that observed for linear carboxylic acids. This may be a consequence of the associated reactivity of fluorinated acids and overall lower degree of functionalization of the surface compared to aliphatic carboxylic acids. Despite this, the fluorinated surfaces remain strongly hydrophobic with apparent contact angles between 145° and 155° depending on the chain used for functionalization and the printed design of the surface. As examples, surfaces functionalized with C_2_F_4_ show, respectively, apparent contact angles of *θ* = 147.0° ± 2.6, *θ* = 156.0° ± 1.1, *θ* = 145.4° ± 2.9 and *θ* = 150.7° ± 0.3 for PVB C_2_F_4_, GVB C_2_F_4_, LMG C_2_F_4_ and NMG C_2_F_4_ surfaces.

As expected, the surfaces functionalized with carboxylic acids bearing polar chains present lower hydrophobicity since the polarity of their side chain affords interaction with water. All the surfaces functionalized with glycolic acid (Gly) present hydrophilic features and no contact angle could be measured due to the spreading of the water into the macro-structures of the surfaces (Figure 9C). While less hydrophobic than surfaces functionalized with aliphatic, fluorinated and perfluorinated carboxylic acids, the Methoxy modified surfaces are surprisingly strongly hydrophobic with contact angles (*θ*) between 130° and 145° depending on the surface macro-structure (PVB Methoxy: *θ* = 146.2° ± 1.8, GVB Methoxy: *θ* = 133.1° ± 0.7, LMG Methoxy: *θ* = 130.9° ± 1.4 and NMG Methoxy: *θ* = 136.2° ± 1.4). Interestingly, it is important to note that all surfaces (except the surfaces modified with Methoxy) present significantly higher hydrophobic feature compared with not functionalized surfaces. This highlights the importance of controlling the surface macroscopic structures and the advantage provided by employing a 3D-printing strategy.

To confirm that the prepared surfaces are parahydrophobic (e.g., display the petal effect), dynamic contact angle measurements were completed [22,23,24]. All the hydrophobic surfaces presented high adhesion with water; a water droplet of 4 µL deposited on any of the prepared hydrophobic functionalized surfaces remains stuck even when the surface is tilted to an inclination angle of 90° (Figure 11).

To show the potential of the developed surfaces for future applications in fog harvesting, it is necessary that the affinity of water to the surface is overcome as the water droplet grows, meaning the water should eventually be released to enable subsequent collection. To investigate this behavior, a water droplet of 100 µL was deposited onto the surfaces and the surfaces were then tilted (Figure 12).

With such a large droplet volume, the surfaces no longer demonstrate parahydrophobic behavior nor having strong adhesion, as all droplets roll off the surface even at small tilt angles (less than 20°). This confirmation allows us to consider that the prepared surfaces will be candidates of interest for future fog harvesting investigations.

## 4. Conclusions

In the present work, we utilize 3D-printing and post-functionalization to create parahydrophobic surfaces. Four surface shapes were designed to increase the specific surface area, then 3D printed and finally post-functionalized with a wide range of carboxylic acids (ten carboxylic acids in total). Surface roughness was characterized after functionalization on the edge of the surface structuration. Roughness measurements were consistent, and one design remain mostly the same regardless the functionalization. Surface morphologies were observed using SEM. These observations on functionalized confirmed the roughness measurements and showed that the functionalized surfaces present thin nano-crystallization as described for such kind of functionalization in the literature [44,45,46]. Most of the prepared surfaces exhibit strong hydrophobicity, with contact angles greater than what has been reported for a simple plate functionalized with a similar procedure [44]. This is particularly evident for GVB and PVB modified with long linear carboxylic acids (C6 to C18). These results show the value of manipulating macroscopic surface design to tailor the surface properties, and also the advantage afforded by employing a 3D-printing strategy. Furthermore, we report the preparation of surfaces with strong hydrophobic features (apparent contact angles greater than 155°) and strong adhesion with water (parahydrophobicity). We confirm that the surface properties allow for the release of water when the volume of the water droplet becomes significantly large. All these results confirm that this elaboration strategy and the prepared surfaces present an important potential for further development in fog harvesting devises.

## Figures and Tables

**Figure 1 biomimetics-07-00122-f001:**
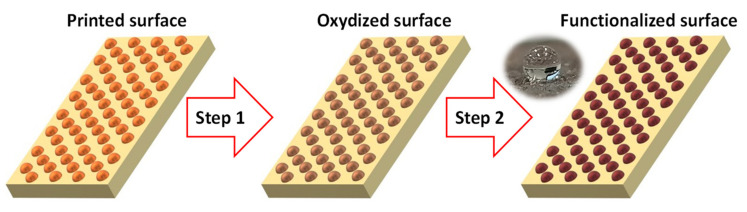
General concept of two step post printing functionalization.

**Figure 2 biomimetics-07-00122-f002:**
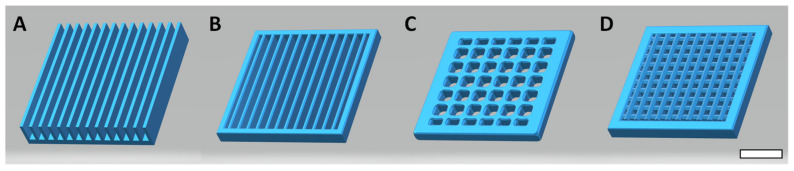
3D structures modelized for printing. (**A**) Plate with vertical blades (PVB), (**B**) Grid with vertical blades (GVB), (**C**) Large mesh grid (GLM) and (**D**) Narrow mesh grid (GNM) (Scale bar = 1 cm).

**Figure 3 biomimetics-07-00122-f003:**
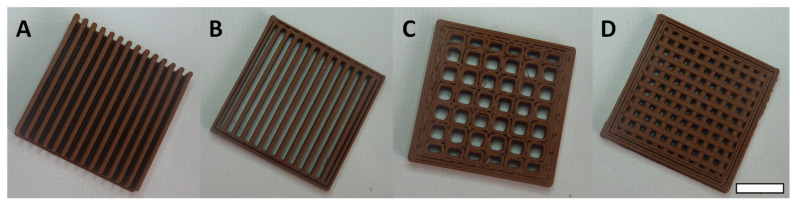
Examples of printed surfaces. (**A**) Plate with vertical blades (PVB), (**B**) Grid with vertical blades (GVB), (**C**) Large mesh grid (GLM) and (**D**) Narrow mesh grid (GNM) (Scale bar = 1 cm).

**Figure 4 biomimetics-07-00122-f004:**
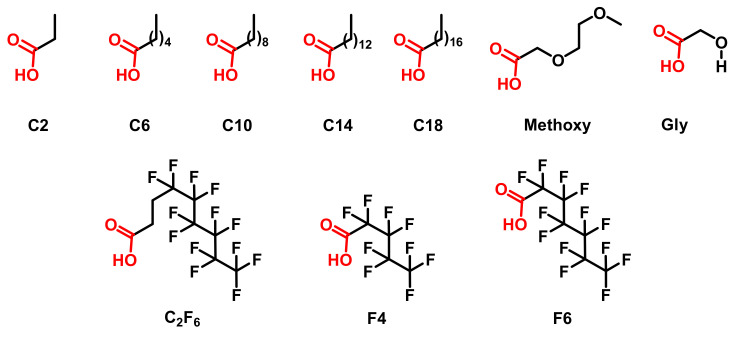
Acids used for post printing functionalization.

**Figure 5 biomimetics-07-00122-f005:**
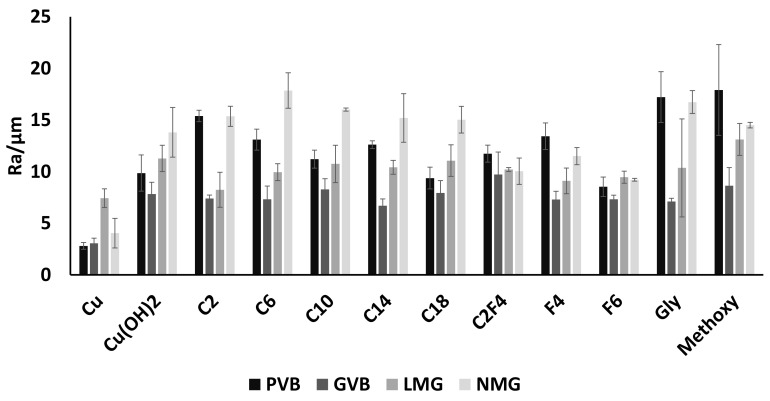
Roughness measurements for printed surfaces for all raw or functionalized surfaces (From dark to clear: PVB, GVB, LMG and NMG).

**Figure 6 biomimetics-07-00122-f006:**
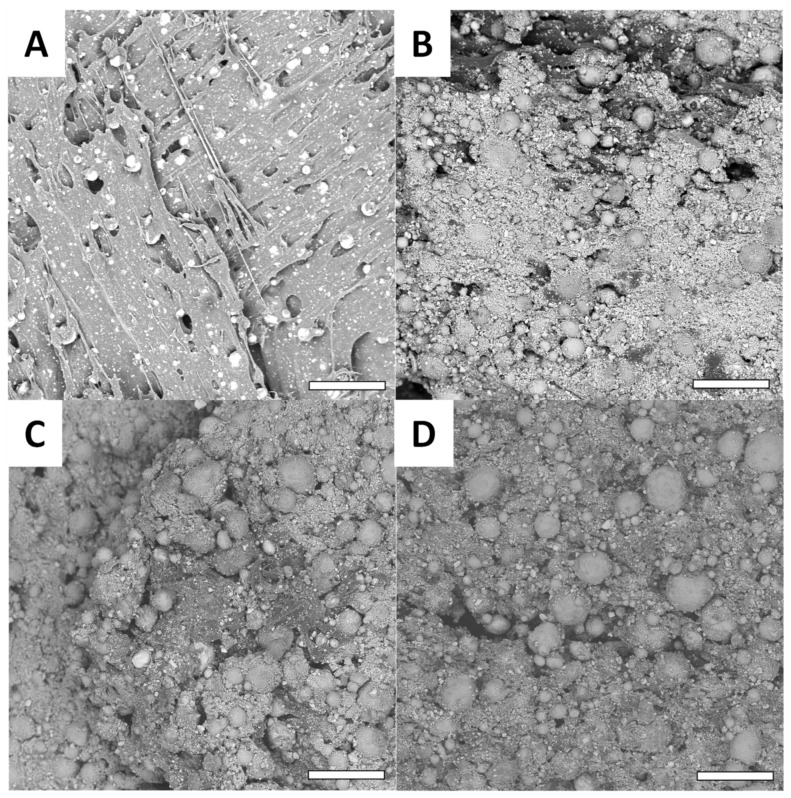
Examples of SEM images with low magnification (scale bar = 100 µm). The presented surfaces are LMG surfaces, (**A**) Cu, (**B**) Cu(OH)_2_, (**C**) Methoxy and (**D**) Gly.

**Figure 7 biomimetics-07-00122-f007:**
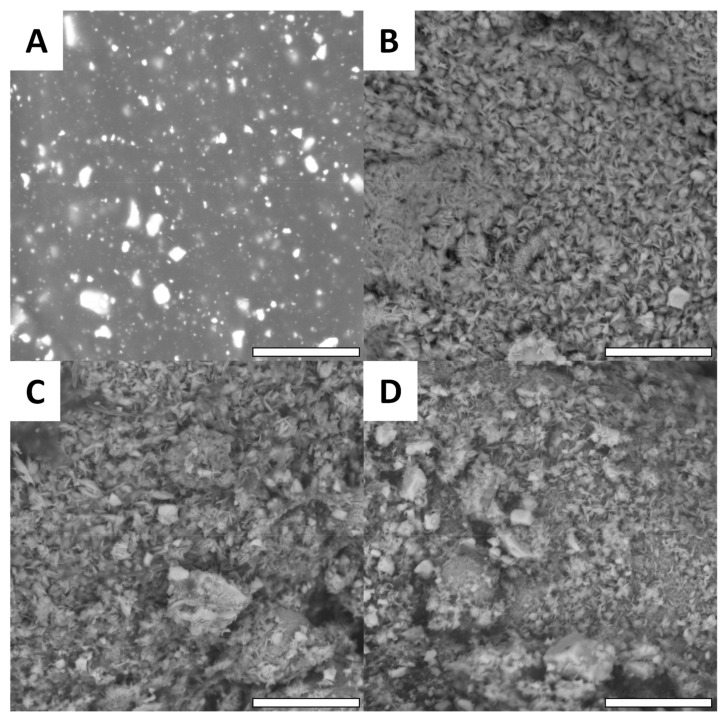
Examples of SEM images with high magnification (scale bar = 8 µm). The presented surfaces are LMG surfaces, (**A**) Cu, (**B**) Cu(OH)_2_, (**C**) Methoxy and (**D**) Gly.

**Figure 8 biomimetics-07-00122-f008:**
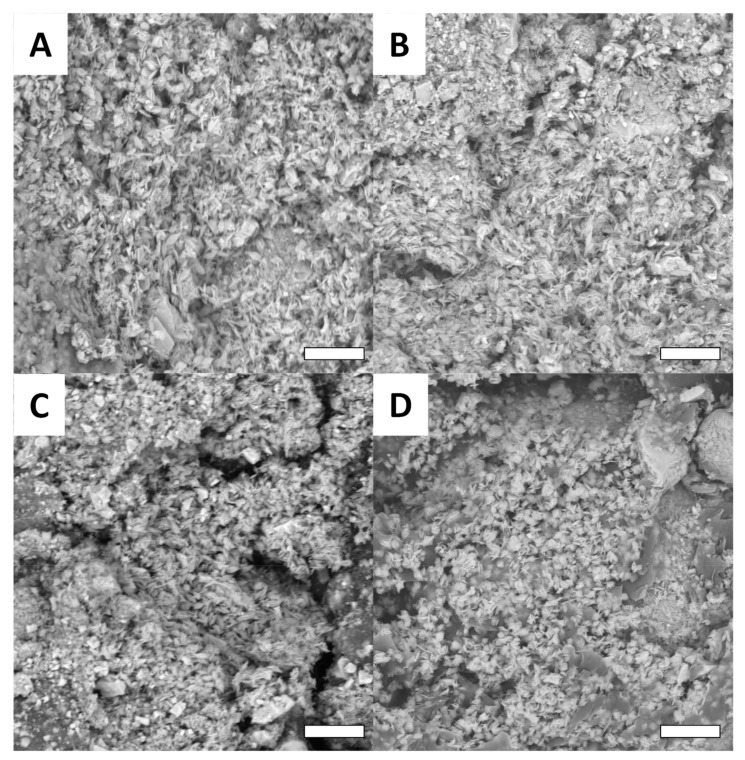
Examples of SEM images for surfaces of different shapes functionalized with C_2_F_4_ (scale bar = 5 µm): PVB (**A**), GVB (**B**), LMG (**C**) and NMG (**D**).

**Figure 9 biomimetics-07-00122-f009:**
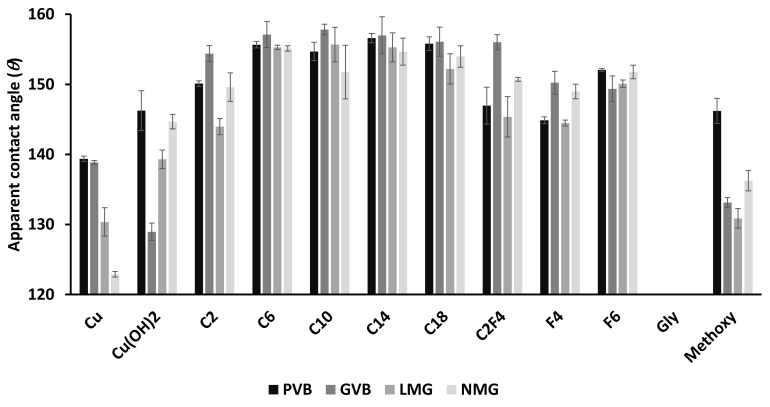
Wettability measurements for printed surfaces for all raw or functionalized surfaces (From dark to clear: PVB, GVB, LMG and NMG).

**Figure 10 biomimetics-07-00122-f010:**
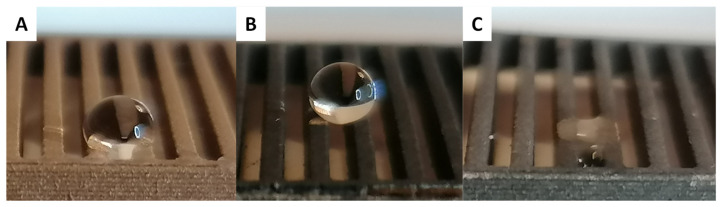
Example of water drop deposed on GVB surfaces. (**A**) Cu, (**B**) C_18_ and (**C**) Gly.

**Figure 11 biomimetics-07-00122-f011:**
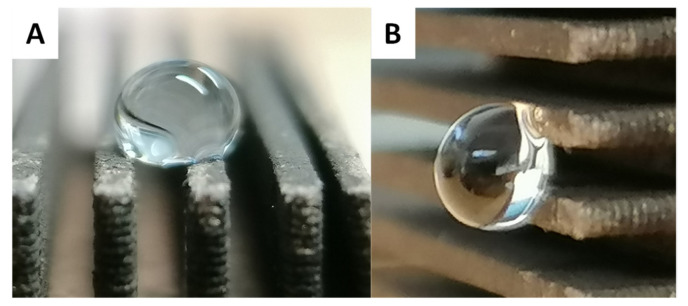
Example of water drop on PVB C_14_. (**A**) On horizontal surface and (**B**) On vertical surface (inclination of 90°).

**Figure 12 biomimetics-07-00122-f012:**
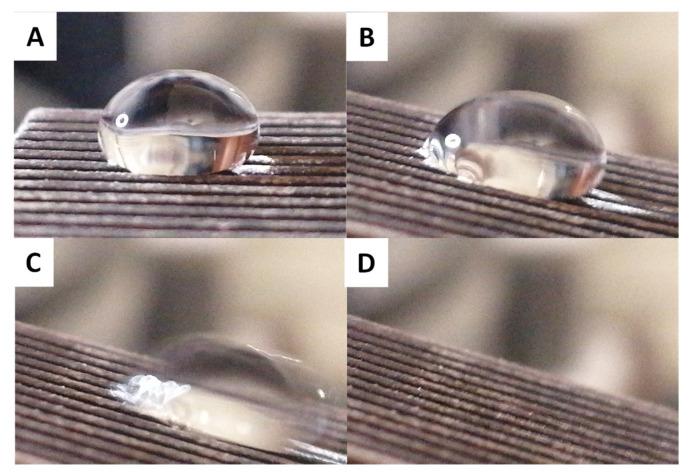
Sliding of 100 µL water drop on PVB C_14_ during the surface is tilted, the tilt angle of the surface increase from (**A**–**D**).

**Table 1 biomimetics-07-00122-t001:** Apparent contact angles (*θ*) measured for printed surfaces for all raw or functionalized surfaces on PVB, GVB, LMG and NMG. The presented values are mean values (more or less the standard deviation over five measurements).

	PVB	GVB	LMG	NMG
Cu	139.4 ± 0.4	138.9 ± 0.3	130.4 ± 2.0	122.9 ± 0.4
Cu(OH)_2_	146.2 ± 2.8	128.9 ± 1.2	139.3 ± 1.3	144.7 ± 1.0
C2	150.1 ± 0.4	154.4 ± 1.1	144.0 ± 1.2	149.6 ± 2.0
C6	155.7 ± 0.5	157.1 ± 1.9	155.3 ± 0.3	155.1 ± 0.4
C10	154.7 ± 1.3	157.8 ± 0.7	155.7 ± 2.5	151.7 ± 3.8
C14	156.6 ± 0.6	157.0 ± 2.6	155.3 ± 2.1	154.7 ± 1.9
C18	155.8 ± 1.0	156.1 ± 2.1	152.2 ± 2.2	154.0 ± 1.5
C2F4	147.0 ± 2.6	156.0 ± 1.1	145.4 ± 2.9	150.7 ± 0.3
F4	144.9 ± 0.5	150.2 ± 1.6	144.5 ± 0.4	149.0 ± 1.0
F6	152.1 ± 0.2	149.4 ± 1.8	150.1 ± 0.5	151.8 ± 1.0
Gly	N/A	N/A	N/A	N/A
Methoxy	146.2 ± 1.8	133.1 ± 0.7	130.9 ± 1.4	136.2 ± 1.4

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
