# Peer review of "Inspired by the Nature: A Post-printed Strategy to Efficiently Elaborate Parahydrophobic Surfaces"

_biomimetics, 2022, doi:10.3390/biomimetics7030122_

Round 1

Reviewer 1 Report

The authors have published similar work last year (Biomimetics 2021, 6, 71). 

At best this ms has incremental novelty over their last publication. 

I recommend rejection.  

Author Response

Reviewer 1

The authors have published similar work last year (Biomimetics 2021, 6, 71). 

At best this ms has incremental novelty over their last publication. 

I recommend rejection.

Author answer: The group admits that this work is incremental compared to the previous work cited above. But our group thinks that there are important new results and the significant improvement of the hydrophobicity of the structured surfaces compared to the flat ones reported previously justifies publication of these new results. In this work, 4 new printed designs are reported with 10 different types of functionalization. As consequence, 40 original surfaces are investigated. These new results highlight significant improvements in terms of wettability and apparent contact angles greater than q = 155 ° are reported in many cases

Reviewer 2 Report

This manuscript presents a nice strategy devoted to water purification. The authors has thoroughly  investigated the influence of various structures, theur surface chemistries and morphologies. They have optimized the design of their structure. 

One minor comment. Can you provide the testing of the water condensation? How your samples will perform in the application? You might try to use it in a humidity box and make some pictures of water droplets condensed on your surfaces and compare them.

Author Response

Reviewer 2

This manuscript presents a nice strategy devoted to water purification. The authors has thoroughly investigated the influence of various structures, their surface chemistries and morphologies. They have optimized the design of their structure. 

Author answer: The group thanks the reviewer for this kind comment.

One minor comment. Can you provide the testing of the water condensation? How your samples will perform in the application? You might try to use it in a humidity box and make some pictures of water droplets condensed on your surfaces and compare them.

Author answer: The group agrees with this comment, however feels that a complete study of the harvesting properties of all surfaces cannot be achieved within the 15-day review period and merits a follow-up study in a separate publication. A complete evaluation of the harvesting capability of our surfaces is planned in the short-term future. This study will include complete evaluation of the harvesting properties for both the smooth and the structured surfaces, with evaluation of the harvesting yield depending on the functionalization nature and the surface design. Water droplet nucleation and the overall harvesting mechanism will also be investigated.

Reviewer 3 Report

This contribution deals with objects showing parahydrophobic qualities, with the aim of devising biomimetic surfaces suitable for harvesting atmospheric water.

The presented prototypes (grids and meshes equipped with parahydrophobic surfaces) are in fact interesting. There are, however, grave problems with the allegedly “biomimetic background” of these objects and also with the evaluation of these structures with respect to fog harvesting.

The authors claim that the “plush-like” hair cover of the biological model, the plant species Echeveria pulvinata, would support the harvest of atmospheric water. It is not clear, however, if and how atmospheric water contributes to the water supply of E. pulvinata.  Firstly, it is not explained how the parahydrophobic hair cover should benefit the harvest of atmospheric water in E. pulvinata. The drop grows on the parahydrophobic surface and finally rolls, and then what? What should be the benefit of a large rolling drop?

Secondly, in many cases (meaning, for many plant species), the adaptive value of such prominent hair covers is not really understood.  The circumstance that plant hairs interact somehow with water does not necessarily mean that these structures are involved in water supply. Rather, various possible benefits and effects are discussed. Brewer and Smith, for instance, suggested possible optical effects (Brewer, C. A., Smith, W. K., & Vogelmann, T. C. (1991). Functional interaction between leaf trichomes, leaf wettability and the optical properties of water droplets. Plant, Cell & Environment, 14(9), 955-962.)

Also, water droplets collecting at a leaf hair cover during fog or dew condensation may reduce transpiration (=evaporation from the leaf when stomata are open for gaining CO2 fo photosynthesis). In this way, leaf hairs might support the water status of a plant indirectly (as discussed by – among others - Brewer and Smith, 1991 or Konrad et al. 2015,  Leaf pubescence as a possibility to increase water use efficiency by promoting condensation. Ecohydrology, 8(3), 480-492). It is unclear, however, whether this may be the case for E. pulvinata, because it is a CAM plant (as is typical for the family of Crassulaceae to which E. pulvinata belongs). This means that gas exchange (= open stomata) occurs during the night when humidity is low anyway.

Also, even if a wet “leaf fur” is beneficial for a plant by reducing transpiration, this does not mean that the leaves are efficient fog or dew harvesters because just a “wet fur” is sufficient.  Also, it is difficult to prove that a certain plant or animal species obtains water supply from fog or dew. There is also a difference between “opportunistic fog harvesters” and species which are specialized on fog and/or dew harvesting (for example, a recent review on plants and animals of the Namib: Mitchell et al. (2020). Fog and fauna of the Namib Desert: past and future. Ecosphere, 11(1), e02996).

Finally, E. pulvinata occurs in mountainous area, at higher altitudes. Alpine plants often show dense trichome covers whose functions are debated. Also, dense trichome covers may affect insect attacks (Amada 2020. Leaf trichomes in Metrosideros polymorpha can contribute to avoiding extra water stress by impeding gall formation. Annals of botany, 125(3), 533-542.).

It is perfectly ok to obtain inspiration from biological observations, also if the functional background of the observed effect is unclear. It is, however, problematic to claim observed effects as “adaptations” to a certain function without biological proof. This should be considered by the authors.

Also, a quantitative and comparative evaluation of the fog-harvesting capacity of the protoypes (page 8) is missing. This is most important for improving fog-collecting systems. Which prototype performs best? And – most important – does a parahydrophic surface or a rough surface increase fog harvesting success (meaning a comparison with structures showing non-parahydrophobic and/or smooth surfaces)?

A number of specific comments are listed below.

P. 1, l. 20: What do the authors mean by “non-liquid”? Presumably, water vapor is meant. “Non-liquid” may also comprise dust and the like.

P. 2, l. 55: “vegetal” Should be “plant”

P. 2, ls. 58: The authors mention the Namib beetle Stenocara gracilipes as a fog-havesting beetle. However, S. gracilipes does not harvest neither fog nor dew (Mitchell et al. 2020). The “real” fog-basking beetle is Onymacris unguicularis, showing a smooth surface (Mitchell et al. 2020 and citations therein).

P. 3, l. 61 and elsewhere: “pine”: spine

P. 3, l. 62: How should the structures increase fog harvesting? By reducing the boundary layer thickness, or any other mechanisms?

P. 3, ls. 66-67: “… the water drop condenses…” Water vapor condenses, a drop is already condensed. Do the authors mean fog or dew harvesting or both?

P. 3, l. 68: “… the droplet is released and the water is collected…” How is the water collected? What happens to the drop, when it is released?

Author Response

Reviewer 3

This contribution deals with objects showing parahydrophobic qualities, with the aim of devising biomimetic surfaces suitable for harvesting atmospheric water.

The presented prototypes (grids and meshes equipped with parahydrophobic surfaces) are in fact interesting. There are, however, grave problems with the allegedly “biomimetic background” of these objects and also with the evaluation of these structures with respect to fog harvesting.

The authors claim that the “plush-like” hair cover of the biological model, the plant species Echeveria pulvinate, would support the harvest of atmospheric water. It is not clear, however, if and how atmospheric water contributes to the water supply of E. pulvinate.  Firstly, it is not explained how the parahydrophobic hair cover should benefit the harvest of atmospheric water in E. pulvinate. The drop grows on the parahydrophobic surface and finally rolls, and then what? What should be the benefit of a large rolling drop?

Author answer: The group thanks the reviewer for this comment. To clarify the biomimetic inspiration for our work, the example of cacti is potentially more easily understood for the reader. We have rewritten the introduction to include a more detailed description of this water harvesting mechanism.

Secondly, in many cases (meaning, for many plant species), the adaptive value of such prominent hair covers is not really understood.  The circumstance that plant hairs interact somehow with water does not necessarily mean that these structures are involved in water supply. Rather, various possible benefits and effects are discussed. Brewer and Smith, for instance, suggested possible optical effects (Brewer, C. A., Smith, W. K., & Vogelmann, T. C. (1991). Functional interaction between leaf trichomes, leaf wettability and the optical properties of water droplets. Plant, Cell & Environment14(9), 955-962.)

Author answer: The group agrees with reviewer, interaction between leaf trichomes and water are more complex. We modified the introduction to clarify this point and added references to support our hypothesis.

Also, water droplets collecting at a leaf hair cover during fog or dew condensation may reduce transpiration (=evaporation from the leaf when stomata are open for gaining CO2 fo photosynthesis). In this way, leaf hairs might support the water status of a plant indirectly (as discussed by – among others - Brewer and Smith, 1991 or Konrad et al. 2015,  Leaf pubescence as a possibility to increase water use efficiency by promoting condensation. Ecohydrology8(3), 480-492). It is unclear, however, whether this may be the case for E. pulvinata, because it is a CAM plant (as is typical for the family of Crassulaceae to which E. pulvinata belongs). This means that gas exchange (= open stomata) occurs during the night when humidity is low anyway.

Also, even if a wet “leaf fur” is beneficial for a plant by reducing transpiration, this does not mean that the leaves are efficient fog or dew harvesters because just a “wet fur” is sufficient.  Also, it is difficult to prove that a certain plant or animal species obtains water supply from fog or dew. There is also a difference between “opportunistic fog harvesters” and species which are specialized on fog and/or dew harvesting (for example, a recent review on plants and animals of the Namib: Mitchell et al. (2020). Fog and fauna of the Namib Desert: past and future. Ecosphere11(1), e02996).

Author answer: The author thanks the reviewer for this comment, in this work, we focus on surface engineering to mimic the property observed in nature. We tried to clarify this point in the introduction.

Finally, E. pulvinata occurs in mountainous area, at higher altitudes. Alpine plants often show dense trichome covers whose functions are debated. Also, dense trichome covers may affect insect attacks (Amada 2020. Leaf trichomes in Metrosideros polymorpha can contribute to avoiding extra water stress by impeding gall formation. Annals of botany125(3), 533-542.).

It is perfectly ok to obtain inspiration from biological observations, also if the functional background of the observed effect is unclear. It is, however, problematic to claim observed effects as “adaptations” to a certain function without biological proof. This should be considered by the authors.

Author answer: The group considered this remark and made significant revisions to the introduction in order to clarify what is known about water harvesting in nature and what is hypothesized. These differences should now be clearer for the reader.

Also, a quantitative and comparative evaluation of the fog-harvesting capacity of the protoypes (page 8) is missing. This is most important for improving fog-collecting systems. Which prototype performs best? And – most important – does a parahydrophic surface or a rough surface increase fog harvesting success (meaning a comparison with structures showing non-parahydrophobic and/or smooth surfaces)?

Author answer: The group agrees with this comment, but a complete study of the harvesting properties of all surfaces cannot be achieved within the review period of 15 days. A complete evaluation of the harvesting capability of our prototype is planned and will be the subject of a future work and separate study. This will include complete evaluation of the harvesting properties for smooth and structured surfaces, with evaluation of the harvesting yield depending on the functionalization and the surface design. Water droplet nucleation and harvesting mechanism will also be investigated.

A number of specific comments are listed below.

  1. 1, l. 20: What do the authors mean by “non-liquid”? Presumably, water vapor is meant. “Non-liquid” may also comprise dust and the like.

Author answer: The text has been modified as suggested to avoid confusion.

  1. 2, l. 55: “vegetal” Should be “plant”

Author answer: This has been modified as suggested by the reviewer.

  1. 2, ls. 58: The authors mention the Namib beetle Stenocara gracilipes as a fog-havesting beetle. However, S. gracilipes does not harvest neither fog nor dew (Mitchell et al. 2020). The “real” fog-basking beetle is Onymacris unguicularis, showing a smooth surface (Mitchell et al. 2020 and citations therein).

Author answer: The authors consider this comment and have modified the text accordingly. In the literature it is not as clear, and some references do describe S. gracilipes as water harvester (see below).

Chakrabarti, U.; Paoli, R.; Chatterjee, S.; Megaridis, C.M. Importance of Body Stance in Fog Droplet Collection by the Namib Desert Beetle. Biomimetics 2019, 4, 59, doi:10.3390/biomimetics4030059.

Nørgaard, T.; Dacke, M. Fog-Basking Behaviour and Water Collection Efficiency in Namib Desert Darkling Beetles. Front Zool 2010, 7, 23, doi:10.1186/1742-9994-7-23.

  1. 3, l. 61 and elsewhere: “pine”: spine

Author answer: It has been changed as suggested.

  1. 3, l. 62: How should the structures increase fog harvesting? By reducing the boundary layer thickness, or any other mechanisms?

Author answer: By increasing the specific surface area, the area available for harvesting is increased and thus the harvesting capability. The authors have added a sentence to clarify this point in the main text.

  1. 3, ls. 66-67: “… the water drop condenses…” Water vapor condenses, a drop is already condensed. Do the authors mean fog or dew harvesting or both?

Author answer: The group thanks the reviewer for this comment. At this point, the prototype has not yet been used for harvesting experiments. Therefore, this answer is theoretical. We aim to work on both in a future study.

  1. 3, l. 68: “… the droplet is released and the water is collected…” How is the water collected? What happens to the drop, when it is released?

Author answer: This aspect of the mechanisms depends on the observed species. A sentence considering the example of cacti species has been added to clarify this point for that specific example.

Reviewer 4 Report

The proposed manuscript within the scope of the Biomimetics journal. Some of the results observations are interesting and the structure of the manuscript was completed. however, if the paper can be improved in the following areas, it would add more value to the readers:

1.     In Abstract and conclusion, please highlight the best functionalization parameters and structure shape for obtaining the hydrophobic surface.

2.     The roughness in four structures after functionalized by C2 to F6 have large variation, but the contact angle is very close, please describe it.

Author Response

Reviewer 4

The proposed manuscript within the scope of the Biomimetics journal. Some of the results observations are interesting and the structure of the manuscript was completed. however, if the paper can be improved in the following areas, it would add more value to the readers:

  1. In Abstract and conclusion, please highlight the best functionalization parameters and structure shape for obtaining the hydrophobic surface.

Author answer: The abstract and conclusion have been modified accordingly based on the reviewer suggestion.

  1. The roughness in four structures after functionalized by C2 to F6 have large variation, but the contact angle is very close, please describe it.

Author answer: Discussion on this point has been added in the main text to address this comment.

Round 2

Reviewer 3 Report

The manuscript was improved markedly by the authors. There are some few comments left.

 The main issue is that it is still not yet clear whether the presented surfaces are applicable for fog or dew harvesting or both. Also, dew and fog are still confused in the manuscript. Both kinds of atmospheric water are fundamentally different and so are the methods of harvesting them. Therefore, clarity is necessary.

 The following text parts are ambiguous on fog and dew and/or fog and dew harvesting.

 P. 1, ls. 19-21: “Among the developed strategies, fog harvesting, which takes advantage of atmospheric water vapor to provide potable water, is a solution of interest due to it’s potential in sustainable development.”

 Fog consists of tiny droplets, it is not water vapor.

 P. 2/3, ls. 50-54: “Among the alternative solutions, fog harvesting is of particular interest.[6] The concept of this  strategy is to condense water present in the atmosphere. This strategy is particularly interesting in desert countries with high hygrometric levels, such as Chile where fog harvesting is already commonly used for irrigation.”

 Since fog consists of water droplets, condensation is not necessary for fog harvesting. Dew harvesting requires condensation.

 P. 3, ls. 59 – 65: “As one example, the  Namib desert beetle (Onymacris unguicularis) traps water on its legs and body.[11–15] Similarly, cacti can collect water with their spines and hairy plants can condense dew from the air to collect water and decrease transpiration.[16–20]. Considering cacti examples, the capacity to harvest water is correlated with two parameters. The first of those parameters is the macrostructures present on each plant that increases the surface area available for water harvesting, mainly spines for cacti.”

 O. onymacris is a fog harvester. With respect to cacti, the authors suggest dew condensation.

 P. 3, ls. 69-71: “These two parameters contribute to the overall mechanism of fog harvesting, which can be briefly  described as follows. Due to the high affinity between the surface of the spines and water, a water droplet will condense and accumulate on these structures.”

 Here, fog and dew are confused, and on the next page (l. 76), the text is again on fog harvesting which is again mentioned on pages 19 (l. 302), and 20 (l. 312).

 Particularly dew harvesting shows a low yield. In contrast, yield of fog harvesting depends substantially on the architecture of harvesting devices (mostly meshes) and environmental conditions, as well as on fog properties (mainly density and size of fog droplets) as well as wind speed and direction with respect to the collector (principles of particle capture are essential):

 See for instance: de Dios Rivera, J. (2011). Aerodynamic collection efficiency of fog water collectors. Atmospheric Research, 102(3), 335-342.

 The authors should therefore be specific in their manuscript on the differences between both types of atmospheric water.

 P. 19, l. 294: “… are parahydrophobic like the E. pulvinata surfaces…” This is the first time, E. pulvinata is mentioned. Probably, it can be cancelled completely from the manuscript.

 A final note on fog harvesting by Namib beetles. In their response, the authors remark that some references do describe S. gracilipes as water harvester and cite Nørgaard & Dacke (2019) and Chakrabarti et al. (2019). Neither study confirms S. gracilipes as water harvester.

 Chakrabarti et al. (2019): This study considers only O. unguicularis.

 Nørgaard, T., & Dacke, M. (2010): This study confirms extensive field observations providing no evidence for fog harvesting of the considered beetles with the exception of O. unguicularis. Excerpt from the conclusions: ”In accordance with earlier reports from the field, we find that O. unguicularis is the only one of our four model beetles that assumes a head standing fog-basking stance in a low temperature environment with artificially produced fog.“

Author Response

The manuscript was improved markedly by the authors. There are some few comments left.

 The main issue is that it is still not yet clear whether the presented surfaces are applicable for fog or dew harvesting or both. Also, dew and fog are still confused in the manuscript. Both kinds of atmospheric water are fundamentally different and so are the methods of harvesting them. Therefore, clarity is necessary.

Author Answer: The authors sincerly thank the reviewer for time and comments that help to improve this article.

 The following text parts are ambiguous on fog and dew and/or fog and dew harvesting.

  1. 1, ls. 19-21: “Among the developed strategies, fog harvesting, which takes advantage of atmospheric water vapor to provide potable water, is a solution of interest due to it’s potential in sustainable development.”

 Fog consists of tiny droplets, it is not water vapor.

Author Answer: The sentence has been modified to be more understandable.

  1. 2/3, ls. 50-54: “Among the alternative solutions, fog harvesting is of particular interest.[6] The concept of this  strategy is to condense water present in the atmosphere. This strategy is particularly interesting in desert countries with high hygrometric levels, such as Chile where fog harvesting is already commonly used for irrigation.”

 Since fog consists of water droplets, condensation is not necessary for fog harvesting. Dew harvesting requires condensation.

Author Answer: The sentence has been modified to be more understandable.

  1. 3, ls. 59 – 65: “As one example, the  Namib desert beetle (Onymacris unguicularis) traps water on its legs and body.[11–15] Similarly, cacti can collect water with their spines and hairy plants can condense dew from the air to collect water and decrease transpiration.[16–20]. Considering cacti examples, the capacity to harvest water is correlated with two parameters. The first of those parameters is the macrostructures present on each plant that increases the surface area available for water harvesting, mainly spines for cacti.”
  2. onymacris is a fog harvester. With respect to cacti, the authors suggest dew condensation.

Author Answer: From literature, fog harvesting is also reported for cacti, of course it depend on the cacti species.. For example: O. microdasys is reported as fog harvester in Ju, J.; Bai, H.; Zheng, Y.; Zhao, T.; Fang, R.; Jiang, L. A Multi-Structural and Multi-Functional Integrated Fog Collection System in Cactus. Nat Commun 2012, 3, 1247, doi:10.1038/ncomms2253.

  1. 3, ls. 69-71: “These two parameters contribute to the overall mechanism of fog harvesting, which can be briefly described as follows. Due to the high affinity between the surface of the spines and water, a water droplet will condense and accumulate on these structures.”

 Here, fog and dew are confused, and on the next page (l. 76), the text is again on fog harvesting which is again mentioned on pages 19 (l. 302), and 20 (l. 312).

 Particularly dew harvesting shows a low yield. In contrast, yield of fog harvesting depends substantially on the architecture of harvesting devices (mostly meshes) and environmental conditions, as well as on fog properties (mainly density and size of fog droplets) as well as wind speed and direction with respect to the collector (principles of particle capture are essential):

 See for instance: de Dios Rivera, J. (2011). Aerodynamic collection efficiency of fog water collectors. Atmospheric Research102(3), 335-342.

 The authors should therefore be specific in their manuscript on the differences between both types of atmospheric water.

Author Answer: The author modified this sentence to focus on fog harvesting and avoid confusion.

  1. 19, l. 294: “… are parahydrophobic like the E. pulvinata surfaces…” This is the first time, E. pulvinata is mentioned. Probably, it can be cancelled completely from the manuscript.

Author Answer: As suggested, E. pulvinate has been canceled.

 A final note on fog harvesting by Namib beetles. In their response, the authors remark that some references do describe S. gracilipes as water harvester and cite Nørgaard & Dacke (2019) and Chakrabarti et al. (2019). Neither study confirms S. gracilipes as water harvester.

 Chakrabarti et al. (2019): This study considers only O. unguicularis.

 Nørgaard, T., & Dacke, M. (2010): This study confirms extensive field observations providing no evidence for fog harvesting of the considered beetles with the exception of O. unguicularis. Excerpt from the conclusions: ”In accordance with earlier reports from the field, we find that O. unguicularis is the only one of our four model beetles that assumes a head standing fog-basking stance in a low temperature environment with artificially produced fog.“

Author Answer: The author thank the reviewer for this remark and take good note of it.